# Self-Supervised Image Restoration with Blurry and Noisy Pairs

**Zhilu Zhang**[1]**, Rongjian Xu**[1]**, Ming Liu**[1]**, Zifei Yan**[1,*]**, Wangmeng Zuo**[1,2]
[1] Harbin Institute of Technology, Harbin, China;
[2] Peng Cheng Laboratory, China
cszlzhang@outlook.com, ronjon.xu@gmail.com,
csmliu@outlook.com, yanzifei@hit.edu.cn, wmzuo@hit.edu.cn

## Abstract

When taking photos under an environment with insufficient light, the exposure time and the sensor gain usually require to be carefully chosen to obtain images with satisfying visual quality. For example, the images with high ISO usually have inescapable noise, while the long-exposure ones may be blurry due to camera shake or object motion. Existing solutions generally suggest to seek a balance between noise and blur, and learn denoising or deblurring models under either full- or self-supervision. However, the real-world training pairs are difficult to collect, and the self-supervised methods merely rely on blurry or noisy images are limited in performance. In this work, we tackle this problem by jointly leveraging the short-exposure noisy image and the long-exposure blurry image for better image restoration. Such setting is practically feasible due to that short-exposure and long-exposure images can be either acquired by two individual cameras or synthesized by a long burst of images. Moreover, the short-exposure images are hardly blurry, and the long-exposure ones have negligible noise. Their complementarity makes it feasible to learn restoration model in a self-supervised manner. Specifically, the noisy images can be used as the supervision information for deblurring, while the sharp areas in the blurry images can be utilized as the auxiliary supervision information for self-supervised denoising. By learning in a collaborative manner, the deblurring and denoising tasks in our method can benefit each other. Experiments on synthetic and real-world images show the effectiveness and practicality of the proposed method. Codes are available at https://github.com/cszhilu1998/SelfIR.

## 1 Introduction

It is a common yet challenging task to acquire visually appealing photos with appropriate brightness under a low light environment. Traditional ways to increase the image brightness include enlarging the aperture, adopting a higher ISO, and reducing the shutter speed (*i.e.*, lengthening exposure time). As for smartphone cameras with fixed aperture, brightness can only be adjusted by setting the sensor gain (*i.e.*, ISO) and exposure time. Nonetheless, they are negatively correlated to maintain the appropriate brightness level of the image, *i.e.*, the shorter-exposure image generally adopts a higher ISO, while the longer-exposure image usually has a lower ISO. Moreover, high ISO configuration introduces inevitable and complex noise due to the limited photon amount and the process of camera image signal processing (ISP) pipeline, while long-exposure is prone to produce blurry images due to the camera shake and scene variations. Consequently, photographers have to make a compromise between noise and blur.

---

[*]Correspondence author.

36th Conference on Neural Information Processing Systems (NeurIPS 2022).

Recent advances in image restoration make it possible to further improve the visual quality of the acquired low light images by leveraging deep image denoising or deblurring networks. Taking supervised denoising as an example, synthetic or real-world noisy-clean image pairs are required to train the deep networks [9, 18, 47–49, 52, 53]. However, the models trained with synthetic training pairs are hard to generalize to real noisy images, and the real-world clean reference images are usually obtained by averaging hundreds of noisy ones [1] or with complicated capturing and processing procedure [29], making the collection of large scale real-world training datasets laborious, expensive, and time-consuming. Such problems greatly limit the deployment of models on more devices with different noise distributions. Alternatively, a surge of self-supervised image denoising methods [11, 13, 15–17, 26, 27, 37, 39, 41, 43, 55, 57] have been developed to avoid the collection of ground-truth (GT) training images, yet are limited in handling complex real-world image noise. Another possible solution is to perform motion deblurring on long-exposure images. Early explorations [8, 20, 31, 34, 35, 45] are mainly given to spatially uniform deblurring caused by camera motion. Recently, the proposal of non-uniform datasets (*e.g.*, GoPro [25], REDS [24], HIDE [36], RealBlur [32], *etc.*) has greatly boosted the research of deblurring in more practical scenes involving both camera shake and object motion [6, 25, 38, 50, 51].

In this paper, we suggest to improve low light imaging by jointly leveraging the short-exposure noisy and long-exposure blurry images. First, such setting is practically feasible. For example, multiple cameras have been equipped in modern smartphones, which can be designed to acquire short-exposure and long-exposure images, respectively. Moreover, one can also synthesize a pair of blurry and noisy images from a long burst of images captured by a camera. Second, the noisy and blurry images convey complementary information, which is beneficial to improve restoration performance and makes self-supervised image restoration (SelfIR) possible. We note that several methods [5, 14, 23, 46] have been suggested to combine the blurry image with their noisy counterpart for better image restoration, yet it remains uninvestigated under the self-supervised regime.

We further present a SelfIR model with blurry and noisy pairs. Even though the blurry and noisy images are both *disturbed*, the short-exposure images taken with high ISO are *hardly blurry*, while the long-exposure images taken with low ISO are generally near *noise-free*. Thus, the long-exposure and short-exposure images can be used to provide some supervision information for each other. On the one hand, the noisy image can serve as an alternative of sharp image to supervise deblurring with negligible performance degradation. On the other hand, the static regions in long-exposure images are noise-free and sharp, which in turn can provide auxiliary supervision information for image denoising. Taking these two aspects into account, we present a collaborative learning (co-learning) method termed SelfIR for deblurring and denoising, which is effective in leveraging the complementary information of long- and short-exposure images and can be learned in a self-supervised manner.

Extensive experiments on synthetic data are conducted to evaluate our SelfIR. Both quantitative and qualitative results show that SelfIR outperforms the state-of-the-art self-supervised denoising methods, as well as the supervised denoising and deblurring counterparts. To further verify the practicality of our SelfIR model, we have also collected a set of 61 real-world blurry and noisy pairs using smartphones. Since there are no corresponding ground-truth images for calculating full-reference image quality assessment (IQA) metrics, we evaluate the restoration results using no-reference IQA metrics. The results show that our method also performs favorably against the competing methods.

To sum up, the main contributions of this work include:

- We take a step forward in leveraging blurry and noisy image pairs for image restoration. Going beyond leveraging their complementarity in improving restoration performance, we show that it can also be utilized for self-supervised learning of the restoration model.

- A self-supervised image restoration model (SelfIR) is proposed, where short-exposure images serve as supervision for the corresponding deblurring task, while the sharp regions in long-exposure images provide auxiliary supervision for self-supervised denoising.

- Extensive experiments on both synthetic and real-world image pairs show that our SelfIR performs favorably against the state-of-the-art self-supervised denoising methods, as well as the baseline supervised deblurring and denoising methods.

## 2 Related Work

In this section, we briefly review burst image denoising and deblurring, as well as self-supervised image denoising and deblurring methods. In addition, we recommend [7] for a comprehensive introduction to the relevant mobile computational photography.

**Burst Image Denoising and Deblurring.** In comparison with a single image, burst images can provide more information that is beneficial for image restoration. Hasinoff *et al.* [10] utilize an FFT-based alignment algorithm and a hybrid 2D/3D Wiener filter to denoise and merge a burst of underexposed frames for low-light photography. KPN [21] predicts spatially variant kernels for every burst noisy image to merge them. BPN [44] extends the KPN method with a basis prediction network and achieves larger denoising kernels under certain computing resource constraints. Aittala *et al.* [2] take both noise and blur into account, and restore sharp and noise-free images from burst images in an order-independent manner.

**Self-Supervised Image Denoising and Deblurring.** Recently, self-supervised learning has drawn upsurging attention in low-level vision. DIP [39] utilizes the image prior implicitly captured by the network structure to repair corrupted images. SelfDeblur [30] respectively models the deep priors of clear image and blur kernel for self-supervised deblurring. For these methods, the networks are required to re-train from scratch for each test image, which is less efficient, especially for mobile or edge devices. Noise2Noise [17] demonstrates that noisy pairs with mutually independent noise can be used to train a denoising network, opening the door to self-supervised denoising. Neighbor2Neighbor [11] utilizes a random neighbor sub-sampler to generate the training pairs from noisy images themselves. In addition, some works [13, 15, 16, 41, 43] elaborately design blind-spot networks to avoid learning the identity mapping for self-supervised denoising. However, the self-supervised denoising methods are limited in handling complex image noise. In this work, we utilize the complementarity of long-exposure blurry and short-exposure noisy images for better self-supervised image restoration.

## 3 Proposed Method

In this section, we first show the feasibility of taking noisy images as the supervision of deblurring. Then, we introduce the sharp area detection method in long-exposure images and auxiliary loss for self-supervised denoising. Finally, we present the proposed co-learning framework SelfIR.

### 3.1 Deblurring with Noisy Image

When taking long-exposure photos, the shake of the camera and the motion of objects usually lead to a blurry image $\mathbf{I}_\mathcal{B}$, which can be formulated by,

$$\mathbf{I}_\mathcal{B} = \mathcal{K}(\mathbf{I}) + \mathbf{N}_\mathcal{B}, \tag{1}$$

where $\mathbf{I}$ is the latent clear image, $\mathcal{K}$ denotes the blur process with non-uniform kernels, $\mathbf{N}_\mathcal{B}$ represents the low-intensity noise. Existing supervised deblurring methods generally utilize a deep neural network (denoted by $\mathcal{D}_\mathcal{B}$) to estimate $\mathbf{I}$ from $\mathbf{I}_\mathcal{B}$. For training the parameters of $\mathcal{D}_\mathcal{B}$, which is denoted by $\Theta_\mathcal{B}$, the optimization objective can be defined by,

$$\Theta_\mathcal{B}^* = \arg\min_{\Theta_\mathcal{B}} \mathbb{E}_{\mathbf{I}_\mathcal{B}, \mathbf{I}} \left[ \mathcal{L}\left(\mathcal{D}_\mathcal{B}(\mathbf{I}_\mathcal{B}; \Theta_\mathcal{B}), \mathbf{I}\right) \right], \tag{2}$$

where $\mathcal{L}$ denotes the loss functions for supervised learning. However, collecting clear images is troublesome in real-world scenes. Inspired by Noise2Noise [17], we show that the noisy short-exposure image can be a substitution of the latent clear image to supervise the task of deblurring.

When taking short-exposure photos under low light environment, the limited photon amount and inherent defects of camera ISP make the images noisy (denoted by $\mathbf{I}_\mathcal{N}$), which can be formulated by,

$$\mathbf{I}_\mathcal{N} = \mathbf{I} + \mathbf{N}_\mathcal{N}, \tag{3}$$

where $\mathbf{N}_\mathcal{N}$ represents the noise (with much higher intensity than $\mathbf{N}_\mathcal{B}$). When using the noisy image $\mathbf{I}_\mathcal{N}$ as the supervision of deblurring, the optimization of $\Theta_\mathcal{B}$ can be expressed as,

$$\Theta_\mathcal{B}^* = \arg\min_{\Theta_\mathcal{B}} \mathbb{E}_{\mathbf{I}_\mathcal{B}, \mathbf{I}_\mathcal{N}} \left[ \mathcal{L}\left(\mathcal{D}_\mathcal{B}(\mathbf{I}_\mathcal{B}; \Theta_\mathcal{B}), \mathbf{I}_\mathcal{N}\right) \right] = \arg\min_{\Theta_\mathcal{B}} \mathbb{E}_{\mathbf{I}_\mathcal{B}} \left[ \mathbb{E}_{\mathbf{I}_\mathcal{N}|\mathbf{I}_\mathcal{B}} \left[ \mathcal{L}\left(\mathcal{D}_\mathcal{B}(\mathbf{I}_\mathcal{B}; \Theta_\mathcal{B}), \mathbf{I}_\mathcal{N}\right) \right] \right]. \tag{4}$$

Suppose that the loss function $\mathcal{L}$ in Eqn. (4) is the $\ell_2$ loss, we have,

$$
\begin{aligned}
\mathbb{E}_{\mathbf{I}_{\mathcal{N}}|\mathbf{I}_{\mathcal{B}}}\left[\mathcal{L}\left(\mathcal{D}_{\mathcal{B}}(\mathbf{I}_{\mathcal{B}};\Theta_{\mathcal{B}}),\mathbf{I}_{\mathcal{N}}\right)\right] &= \mathbb{E}_{\mathbf{I}_{\mathcal{N}}|\mathbf{I}_{\mathcal{B}}}\left[\|\mathcal{D}_{\mathcal{B}}(\mathbf{I}_{\mathcal{B}};\Theta_{\mathcal{B}})-\mathbf{I}_{\mathcal{N}}\|_2^2\right] \\
&= \mathbb{E}_{\mathbf{I},\mathbf{N}_{\mathcal{N}}|\mathbf{I}_{\mathcal{B}}}\left[\|\mathcal{D}_{\mathcal{B}}(\mathbf{I}_{\mathcal{B}};\Theta_{\mathcal{B}})-(\mathbf{I}+\mathbf{N}_{\mathcal{N}})\|_2^2\right] \\
&= \mathbb{E}_{\mathbf{I}|\mathbf{I}_{\mathcal{B}}}\left[\|\mathcal{D}_{\mathcal{B}}(\mathbf{I}_{\mathcal{B}};\Theta_{\mathcal{B}})-\mathbf{I}\|_2^2\right]- \\
&\quad 2\mathbb{E}_{\mathbf{I},\mathbf{N}_{\mathcal{N}}|\mathbf{I}_{\mathcal{B}}}\left[(\mathcal{D}_{\mathcal{B}}(\mathbf{I}_{\mathcal{B}};\Theta_{\mathcal{B}})-\mathbf{I})^{\top}\mathbf{N}_{\mathcal{N}}\right]+ \\
&\quad \mathbb{E}_{\mathbf{N}_{\mathcal{N}}|\mathbf{I}_{\mathcal{B}}}\left[\|\mathbf{N}_{\mathcal{N}}\|_2^2\right],
\end{aligned}
\tag{5}
$$

where $\mathbb{E}_{\mathbf{N}_{\mathcal{N}}|\mathbf{I}_{\mathcal{B}}}\left[\|\mathbf{N}_{\mathcal{N}}\|_2^2\right]$ can be regarded as a constant and be safely discarded from Eqn. (5). Further, assume that $\mathbf{N}_{\mathcal{N}}$ is zero-mean, $\mathbf{N}_{\mathcal{N}}$ and $\mathbf{I}$ are independent, we can get,

$$
\mathbb{E}_{\mathbf{I},\mathbf{N}_{\mathcal{N}}|\mathbf{I}_{\mathcal{B}}}\left[(\mathcal{D}_{\mathcal{B}}(\mathbf{I}_{\mathcal{B}};\Theta_{\mathcal{B}})-\mathbf{I})^{\top}\mathbf{N}_{\mathcal{N}}\right]=0.
\tag{6}
$$

In this case, the optimal solution $\Theta_{\mathcal{B}}^*$ in Eqn. (4) and that in Eqn. (2) are the same. Thus, it is feasible to utilize noisy short-exposure images instead of clear ones as the supervision of deblurring.

## 3.2 Denoising with Long-Exposure Image

Self-supervised denoising makes it possible to remove the noise without clean image supervision, but may give rise to obvious performance degradation, especially when handling complex real-world noises. Here we propose to alleviate this problem by introducing some extra supervision from the long-exposure counterpart. Obviously, taking the whole long-exposure image as supervision will bring adverse effects, making the results to be blurry. Nonetheless, it is worth noting that, the blur process $\mathcal{K}$ in Eqn. (1) is generally non-uniform, and sometimes not all areas are blurry. As shown in Fig. 1(d), there exist some approximately sharp regions in the long-exposure image, which can provide partial supervision information that benefits self-supervised denoising.

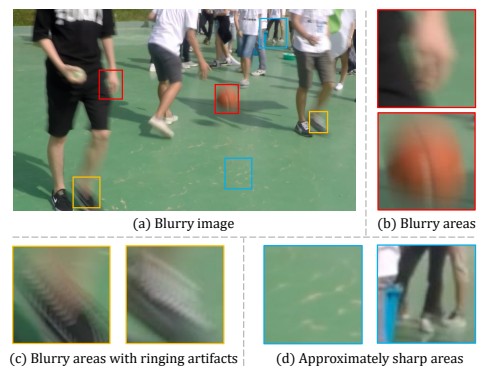

(a) Blurry image    (b) Blurry areas

(c) Blurry areas with ringing artifacts    (d) Approximately sharp areas

Figure 1: A blurry image example where the blur is non-uniform in (a). It includes some common blurry areas (b), severe blurry areas with ringing artifacts (c), and some approximately sharp areas (d).

Therefore, it is crucial to pinpoint the sharp areas in the long-exposure image. Otherwise, we would prefer to go without sharp areas than accept a shoddy option, as misjudgments of sharp areas will lead to worse denoising results. However, without any discriminative clues, it is very likely to misjudge the sharp areas. Considering that $\mathbf{I}_{\mathcal{N}}$ is nearly non-blurry due to the short exposure time, it may be a reference for sharp area detection. To avoid noise interference, we pre-process $\mathbf{I}_{\mathcal{N}}$ with a self-supervised denoising model, and take the result $\tilde{\mathbf{I}}_{\mathcal{N}}$ to help detect sharp regions in the corresponding blurry image $\mathbf{I}_{\mathcal{B}}$.

Specifically, we first divide $\mathbf{I}_{\mathcal{B}}$ and $\tilde{\mathbf{I}}_{\mathcal{N}}$ into $N$ non-overlapping patches. For each patch pair $\mathbf{I}_{\mathcal{B}}^n$ and $\tilde{\mathbf{I}}_{\mathcal{N}}^n$ ($1 \le n \le N$), our goal is to obtain a mask $m^n \in \{0,1\}$ that indicates whether $\mathbf{I}_{\mathcal{B}}^n$ is a sharp patch. Since $\tilde{\mathbf{I}}_{\mathcal{N}}$ is nearly non-blurry, when some severe motion blurs exist in $\mathbf{I}_{\mathcal{B}}^n$, the difference between $\mathbf{I}_{\mathcal{B}}^n$ and $\tilde{\mathbf{I}}_{\mathcal{N}}^n$ in textures and edges should be evident. Taking the above into account, we adopt a similarity metric $s$ to detect the areas with severe motion blurs, i.e.,

$$
m^n = \mathtt{sgn}(\mathtt{max}(0, s(\mathbf{I}_{\mathcal{B}}^n, \tilde{\mathbf{I}}_{\mathcal{N}}^n)-\epsilon_s)),
\tag{7}
$$

where structural similarity (SSIM) [40] is utilized for the similarity metric $s$, $\epsilon_s$ denotes the threshold, while $\mathtt{max}(a,b)$ and $\mathtt{sgn}(\cdot)$ denote the maximum and sign function, respectively.

However, the initial denoising result $\tilde{\mathbf{I}}_{\mathcal{N}}$ may be over-smooth, in other words, Eqn. (7) may fail when facing some mildly blurred regions in $\mathbf{I}_{\mathcal{B}}^n$. Therefore, we further measure the difference in variance between $\mathbf{I}_{\mathcal{B}}^n$ and $\tilde{\mathbf{I}}_{\mathcal{N}}^n$. When the variance of $\mathbf{I}_{\mathcal{B}}^n$ is greater than that of $\tilde{\mathbf{I}}_{\mathcal{N}}^n$, we consider that $\mathbf{I}_{\mathcal{B}}^n$ is potential to be a sharp patch. It should be noted that the difference in variance is not suitable for

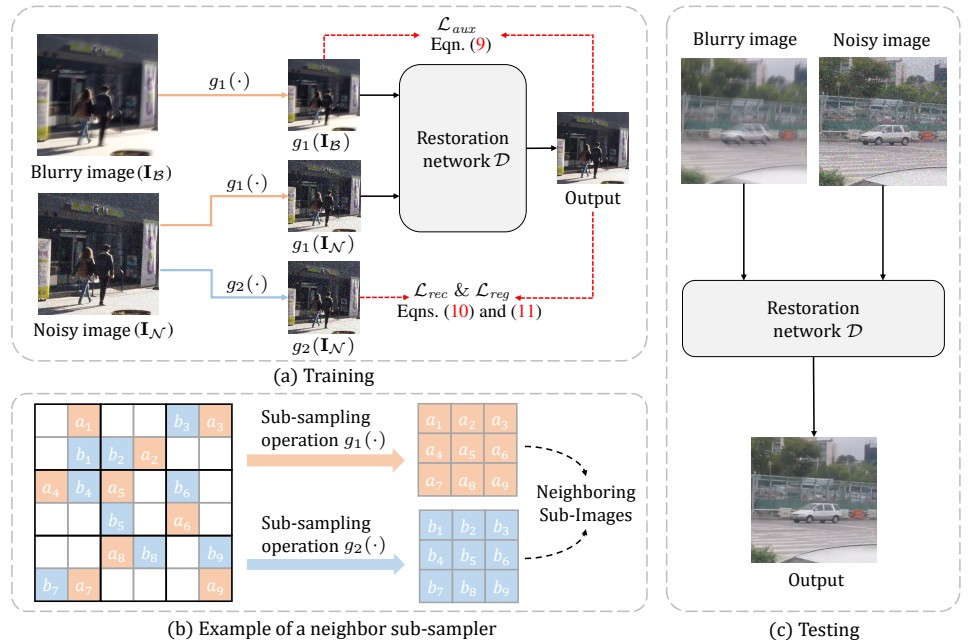

Figure 2: Overview of our proposed SelfIR framework. (a) Training phase of SelfIR. Sub-sampled blurry image $g_1(\mathbf{I}_{\mathcal{B}})$ and noisy image $g_1(\mathbf{I}_{\mathcal{N}})$ are taken as the inputs. $g_2(\mathbf{I}_{\mathcal{N}})$ is used for calculating the reconstruction loss $\mathcal{L}_{rec}$ (see Eqn. (10)) and regularization loss $\mathcal{L}_{reg}$ (see Eqn. (11)), while $g_1(\mathbf{I}_{\mathcal{B}})$ is taken for calculating auxiliary loss (see Eqn. (9)). (b) Example of neighbor sub-sampler. In each $2 \times 2$ cell, two pixels are randomly selected for respectively composing the neighboring sub-images. (c) Testing phase of SelfIR. The blurry and noisy images can be directly taken for restoration.

detecting some severely blurry areas with ringing artifacts (see Fig. 1(c)). In such blurry regions, the variance of $\mathbf{I}_{\mathcal{B}}^n$ may also be greater than that of $\tilde{\mathbf{I}}_{\mathcal{N}}^n$. When synthesizing blurry images, some works [24, 42, 56] remove the artifacts by interpolating the frames before averaging the sharp images. However, the artifacts also exist in real-world blurry images, especially in areas with flickering lights. Therefore, we still take the ringing artifacts into consideration in this work.

As a result, we jointly use the SSIM and variance measure for judging sharp regions, $m^n$ can be formulated as,

$$m^n = \mathtt{sgn}(\max(0, s(\mathbf{I}_{\mathcal{B}}^n, \tilde{\mathbf{I}}_{\mathcal{N}}^n) - \epsilon_s)) * \mathtt{sgn}(\max(0, \mathtt{var}(\mathbf{I}_{\mathcal{B}}^n) - \mathtt{var}(\tilde{\mathbf{I}}_{\mathcal{N}}^n) - \epsilon_v)), \tag{8}$$

where $\mathtt{var}(\cdot)$ and $\epsilon_v$ denote the variance function and the threshold, respectively. $\epsilon_s$ and $\epsilon_v$ are set to 0.99 and 1e-5 when color intensity values of $\mathbf{I}_{\mathcal{B}}^n$ and $\tilde{\mathbf{I}}_{\mathcal{N}}^n$ are normalized to $[0, 1]$. Defining the final output image as $\hat{\mathbf{I}}$, the auxiliary loss function for denoising can be denoted as,

$$\mathcal{L}_{aux}(\hat{\mathbf{I}}, \mathbf{I}_{\mathcal{B}}) = \sum_{n=1}^{N} m^n \|\hat{\mathbf{I}}^n - \mathbf{I}_{\mathcal{B}}^n\|_2^2. \tag{9}$$

### 3.3 Co-Learning of Deblurring and Denoising

As illustrated in Secs. 3.1 and 3.2, the noisy images can provide effective supervision information for deblurring, while the blurry images can also provide auxiliary supervision for self-supervised denoising. In other words, we can learn to deblur or denoise without additional clear and clean ground-truths. However, learning them independently will lead to limited performance. As some works [5, 14, 23, 46] have suggested, better restoration performance can be obtained by incorporating blurry with noisy images. Thus, it should be further considered to achieve self-supervised image restoration that takes two disturbed images as input. In this section, we delicately design the self-supervised model SelfIR that learns from both deblurring and denoising tasks in a collaborative learning manner.

For training SelfIR, it is required to avoid the trivial solution when taking both blurry and noisy images as the input. For example, when taking noisy images for supervising deblurring, the model

may simply output the noisy images instead of learning the deburring results. Fortunately, Neighbor2Neighbor [11] shows the noises in two sub-sampled images (*i.e.*, $g_1(\mathbf{I}_\mathcal{N})$ and $g_2(\mathbf{I}_\mathcal{N})$, see Fig. 2(b)) from noisy image $\mathbf{I}_\mathcal{N}$ are almost independent. Therefore, they can be respectively taken as the input and target for training the denoising subtask in a self-supervised manner. Simultaneously, combining the derivation in Eqn. (5), $g_1(\mathbf{I}_\mathcal{B})$ and $g_2(\mathbf{I}_\mathcal{N})$ can be respectively taken as the input and target to train the deblurring subtask. Therefore, our SelfIR can take a step forward and deliver both $g_1(\mathbf{I}_\mathcal{B})$ and $g_1(\mathbf{I}_\mathcal{N})$ (*i.e.*, $\{g_1(\mathbf{I}_\mathcal{B}, g_1(\mathbf{I}_\mathcal{N}))\}$) into the restoration network $\mathcal{D}$, as shown in Fig. 2(a). The sub-sampled image $g_2(\mathbf{I}_\mathcal{N})$ is taken for calculating reconstruction loss $\mathcal{L}_{rec}$, which can be written as,

$$\mathcal{L}_{rec} = \|\mathcal{D}(g_1(\mathbf{I}_\mathcal{B}), g_1(\mathbf{I}_\mathcal{N})) - g_2(\mathbf{I}_\mathcal{N})\|_2^2. \tag{10}$$

We also calculate regularization loss $\mathcal{L}_{reg}$ similar to [11],

$$\mathcal{L}_{reg} = \|\mathcal{D}(g_1(\mathbf{I}_\mathcal{B}), g_1(\mathbf{I}_\mathcal{N})) - g_2(\mathbf{I}_\mathcal{N}) - (g_1(\hat{\mathcal{D}}(\mathbf{I}_\mathcal{B}, \mathbf{I}_\mathcal{N})) - g_2(\hat{\mathcal{D}}(\mathbf{I}_\mathcal{B}, \mathbf{I}_\mathcal{N})))\|_2^2, \tag{11}$$

where $\hat{\mathcal{D}}$ has same parameters with $\mathcal{D}$ but has no gradient for back-propagation. Moreover, the sharp areas in $g_1(\mathbf{I}_\mathcal{B})$ can provide auxiliary supervision for the model learning and the performance can be further improved. Specifically, we calculate auxiliary loss $\mathcal{L}_{aux}(\mathcal{D}(g_1(\mathbf{I}_\mathcal{B}), g_1(\mathbf{I}_\mathcal{N}))), g_1(\mathbf{I}_\mathcal{B}))$ in Eqn. (9). For obtaining the mask $m^n$, we replace $\mathbf{I}_\mathcal{B}$ and $\tilde{\mathbf{I}}_\mathcal{N}$ in Eqn. (8) with $g_1(\mathbf{I}_\mathcal{B})$ and $\hat{\mathcal{D}}(g_1(\mathbf{I}_\mathcal{B}), g_1(\mathbf{I}_\mathcal{N}))$, respectively. Finally, the learning objective of parameters $\Theta_\mathcal{D}$ with the co-learning manner can be formulated as,

$$\Theta_\mathcal{D}^* = \arg\min_{\Theta_\mathcal{D}} \mathbb{E}_{\mathbf{I}_\mathcal{B}, \mathbf{I}_\mathcal{N}}(\mathcal{L}_{rec} + \lambda_{reg}\mathcal{L}_{reg} + \lambda_{aux}\mathcal{L}_{aux}), \tag{12}$$

where $\lambda_{reg}$ and $\lambda_{aux}$ are hyper-parameters for balancing the loss terms.

## 4 Experiments

In this section, we first describe the dataset configuration and the training / evaluation protocols in detail. Then, both the quantitative and qualitative results on synthetic and real-world images are given for comprehensive evaluation.

### 4.1 Implementation Details

**Synthetic Datasets.** For synthesizing blurry images, early methods [23, 46] convolve sharp images with simulated blur kernels, which is quite different from the real-world blur model. Recently, the GoPro dataset[2] [25] offers a more realistic way to synthesize blurry images, which has been widely adopted for motion deblurring tasks [6, 25, 38, 50, 51]. In the dataset, although slight inherent noise exists in the sharp video frames that are captured by a high-speed camera (*i.e.*, GoPro Hero4 Black), it has little effect on model training and evaluation. We directly regard the sharp images as clean images. The blurry image is generated by averaging consecutive sharp frames, which can better simulate both camera shake and object motion. Thus, we can get the blurry-clear pairs as $\mathbf{I}_\mathcal{B}$ and $\mathbf{I}$, respectively. For synthesizing the noisy image $\mathbf{I}_\mathcal{N}$, we consider three noise distributions: 1) Gaussian noise with $\sigma \in [5/255, 50/255]$, 2) Poisson noise with $\lambda \in [5, 50]$, and 3) sensor noise [4]. Please refer to the supplementary material for the noise formulations. Experiments are conducted in both sRGB and raw-RGB spaces. For the sRGB space, we add Gaussian or Poisson noise to the sRGB sharp images. While for the raw-RGB space, the more complex sensor noise is added to the sharp raw-RGB images, and the blurry images are also converted back to the raw-RGB space through the unprocessing [4] pipeline. Finally, there are 2,103 image pairs for training, and we use the remaining 1,111 pairs for testing.

**Real-world Datasets.** Furthermore, we have also collected 61 real-world blurry and noisy raw-RGB pairs with Huawei P40 smartphones in the professional camera mode. In particular, the blurry images are captured with a low ISO and long exposure time, while the corresponding noisy images are captured with a high ISO and short exposure time in the same scene. Please refer to the supplementary material for detailed collection process. Due to the limited amount of real-world images, we use 30 pairs to fine-tune the model pre-trained on raw-RGB images with synthetic sensor noise, and take the remaining 31 pairs for evaluation.

---

[2]https://seungjunnah.github.io/Datasets/gopro.html

Table 1: Quantitative results on synthetic sRGB images.

| | Method | Gaussian $\sigma \in [5/255, 50/255]$ PSNR↑ / SSIM↑ / LPIPS↓ | Poisson $\lambda \in [5, 50]$ PSNR↑ / SSIM↑ / LPIPS↓ |
|---|---|---|---|
| Supervised Deblurring | Baseline$_{\mathcal{B}}$ | 28.24 / 0.8561 / 0.191 | |
| | DeepDeblur [25] | 30.04 / 0.9015 / 0.133 | |
| Supervised Denoising | Baseline$_{\mathcal{N}}$ | 34.91 / 0.9360 / 0.098 | 33.15 / 0.9225 / 0.126 |
| | DnCNN [52] | 34.63 / 0.9308 / 0.121 | 32.45 / 0.9084 / 0.128 |
| Supervised IR | Baseline$_{\mathcal{R}}$ | 36.15 / 0.9534 / 0.070 | 34.74 / 0.9454 / 0.084 |
| Self-Supervised Denoising | N2N [17] | 34.88 / 0.9354 / 0.100 | 33.09 / 0.9216 / 0.129 |
| | N2V [13] | 33.09 / 0.9180 / 0.115 | 31.81 / 0.8999 / 0.137 |
| | Laine19-mu [15] | 33.61 / 0.9227 / 0.104 | 32.29 / 0.9091 / 0.131 |
| | Laine19-pme [15] | 34.76 / 0.9322 / 0.086 | 32.77 / 0.9147 / 0.116 |
| | DBSN [43] | 33.72 / 0.9224 / 0.111 | 31.46 / 0.8883 / 0.144 |
| | R2R [27] | 33.74 / 0.9223 / 0.100 | 30.05 / 0.7649 / 0.230 |
| | Neighbor2Neighbor [11] | 34.29 / 0.9271 / 0.085 | 32.68 / 0.9160 / 0.111 |
| | Blind2Unblind [41] | 34.69 / 0.9353 / 0.107 | 33.09 / 0.9216 / 0.132 |
| Ours | SelfIR | 35.74 / 0.9499 / 0.076 | 34.27 / 0.9404 / 0.092 |

**Training Details.** Following [11, 15, 17], we adopt a U-Net [33] architecture as our restoration network, where two encoders are respectively deployed to the blurry and noisy input images for better domain-specific feature extraction, and then blurry and noisy features are fused in the decoder part. Detailed architecture is given in the supplementary material. During training, the batch size is set to 16 and the patch size is $128 \times 128$. Adam optimizer [12] with $\beta_1 = 0.9$ and $\beta_2 = 0.999$ is used to train the network for 200 epochs. The learning rate is initially set to $3 \times 10^{-4}$ for synthetic experiments and $1 \times 10^{-4}$ for real-world experiments. And it reduces by half every 50 epochs. For the hyper-parameters in Eqn. (12), $\lambda_{aux}$ is set to 2, $\lambda_{reg}$ is set to 2 and 4 for experiments in sRGB space and raw-RGB space, respectively. All experiments are conducted with PyTorch [28] on an Nvidia GeForce RTX 2080Ti GPU.

**Evaluation Configurations.** We convert all raw-RGB results to sRGB space through post-processing pipeline, and the quantitative metrics are computed in the sRGB space. For the results of synthetic experiments, we take peak signal to noise ratio (PSNR), structural similarity (SSIM) [40] and learned perceptual image patch similarity (LPIPS) [54] as evaluation metrics. For the results of real-world experiments, due to the lack of ground-truth images, we utilize no-reference IQA metrics (*i.e.*, NIQE [22], NRQM [19], and PI [3]) to evaluate the generated images.

## 4.2 Experimental Results in sRGB Space

To assess our proposed SelfIR, we build several baseline methods for comparison, including 1) a supervised deblurring baseline (denoted by Baseline$_{\mathcal{B}}$), 2) a supervised denoising baseline (denoted by Baseline$_{\mathcal{N}}$), 3) a supervised image restoration baseline which takes both blurry and noisy images as input (denoted by Baseline$_{\mathcal{R}}$), and 4) Neighbor2Neighbor [11] since the denoising part of our co-learning framework is based on it. Note that the network structure is not the focus of this paper, so we deploy a simple U-Net [33] architecture for our restoration network $\mathcal{D}$ and the first three baseline methods. Besides, we choose two classical supervised methods (*i.e.*, DeepDeblur [25] and DnCNN [52]) for comparison. Self-supervised denoising methods (*e.g.*, N2N [17], N2V [13], Laine19 [15], DBSN [43], R2R [27], and Blind2Unblind [41]) are also compared. For DeepDeblur [25], we use the officially released model for testing. For other methods, the models are retrained with our synthetic images for a fair comparison.

The quantitative results of synthetic experiments on Gaussian and Poisson noise are given in Tab. 1. One can see that our SelfIR outperforms all other methods except the supervised IR model (Baseline$_{\mathcal{R}}$), which can be seen as the upper limit of the restoration performance with the utilized U-Net architecture and the blurry-noisy pair input. In comparison to the self-supervised denoising baseline Neighbor2Neighbor [11], 1.45 dB and 1.59 dB PSNR gains are obtained by our SelfIR on Gaussian and Poisson noise, respectively. Even compared to the supervised denoising baseline

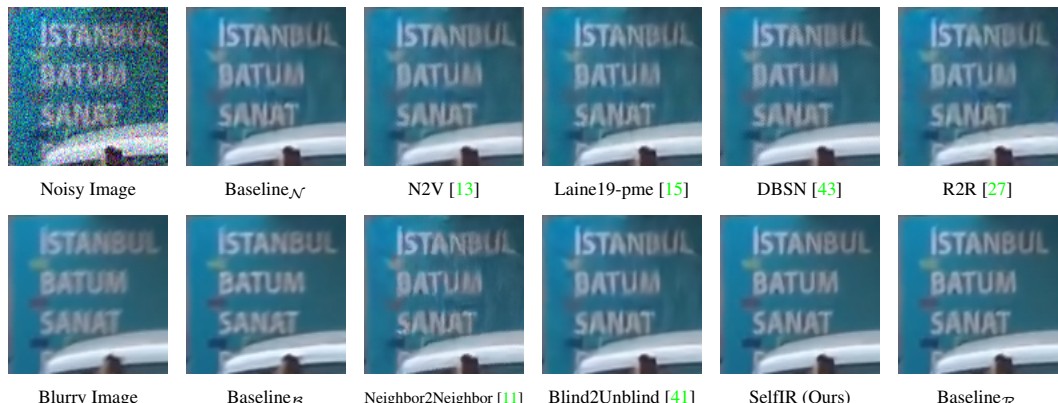

Figure 3: Visual comparison on Gaussian noise. The texts in our result are sharper and clearer. The proposed SelfIR is almost comparable to supervised Baseline$_\mathcal{R}$, which can be seen as the upper limit of the restoration performance with the utilized U-Net architecture and the blurry-noisy pair input.

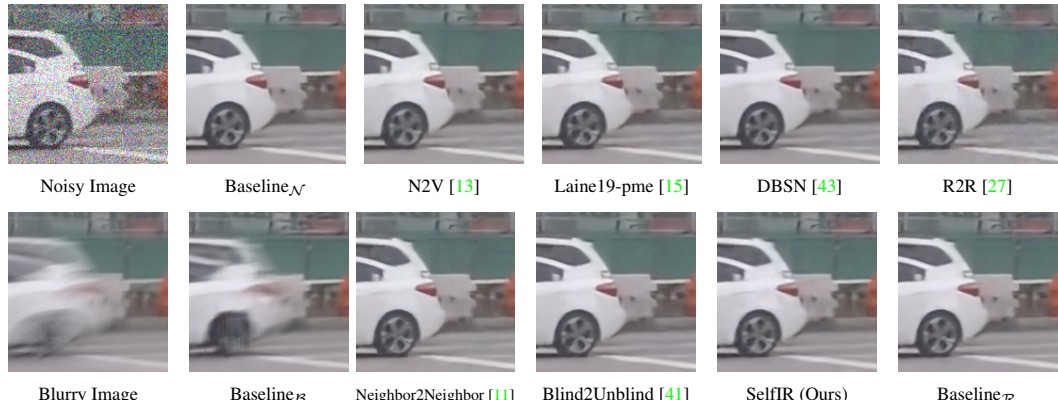

Figure 4: Visual comparison on Poisson noise. In terms of the visual result, SelfIR combines the advantages of supervised denoising Baseline$_\mathcal{N}$ and deblurring Baseline$_\mathcal{B}$.

model (Baseline$_\mathcal{N}$), our method also improves the PSNR index by 0.83 dB and 1.12 dB. Due to the severely ill-posed nature of deblurring, SelfIR brings a PSNR gain by more than 6 dB comparing to the supervised deblurring baseline model (Baseline$_\mathcal{B}$). The results clearly show that the blurry images can indeed provide beneficial features for denoising, and so do the noisy images for deblurring.

The qualitative results are shown in Figs. 3 and 4. One can see that our results are sharper and clearer than other denoising or deblurring methods. In terms of visual quality, the proposed SelfIR is almost comparable to the supervised image restoration model (Baseline$_\mathcal{R}$), and combines the advantages of supervised denoising Baseline$_\mathcal{N}$ and deblurring Baseline$_\mathcal{B}$. More results can be found in the supplemental material.

## 4.3 Experimental Results in Raw-RGB Space

Tab. 2 shows the quantitative results of synthetic and real-world experiments on raw-RGB images. For synthetic experiments with more complex and realistic sensor noise [4], all models are retrained with our synthetic images. Our SelfIR achieves a 1.69 dB PSNR gain in comparison with Neighbor2Neighbor [11]. For evaluation on real-world images, we directly use the models of N2N [17], Laine19-pme [15], R2R [27] and all supervised methods, which are pre-trained for sensor noise. For SelfIR and other self-supervised methods, we fine-tune the pre-trained model on our real-world training set. The PI [3] metric of SelfIR is improved by 0.55 on real-world testing images through fine-tuning. And the results on no-reference IQA show that our method is very competitive in comparison with the competing methods on real-world images. The visual results are given in the supplemental material.

Table 2: Quantitative results on synthetic and real-world raw-RGB images.

| | Method | Sensor Noise [4] PSNR↑ / SSIM↑ / LPIPS↓ | Real-World Images NIQE↓ / NRQM↑ / PI↓ |
|---|---|---|---|
| Supervised Deblurring | Baseline$_\mathcal{B}$ | 28.14 / 0.8547 / 0.162 | 6.26 / 5.04 / 5.62 |
| | DeepDeblur [25] | 29.75 / 0.8881 / 0.115 | 6.76 / 4.78 / 6.00 |
| Supervised Denoising | Baseline$_\mathcal{N}$ | 34.52 / 0.9461 / 0.053 | 5.69 / 4.85 / 5.43 |
| | DnCNN [52] | 33.81 / 0.9325 / 0.076 | 6.05 / 5.10 / 5.48 |
| Supervised IR | Baseline$_\mathcal{R}$ | 36.10 / 0.9574 / 0.035 | 5.54 / 5.14 / 5.18 |
| Self-Supervised Denoising | N2N [17] | 34.67 / 0.9472 / 0.053 | 6.10 / 4.93 / 5.59 |
| | N2V [13] | 31.39 / 0.9227 / 0.076 | 5.82 / 5.52 / 5.17 |
| | Laine19-mu [15] | 32.74 / 0.9304 / 0.073 | 5.87 / 5.67 / 5.10 |
| | Laine19-pme [15] | 33.28 / 0.9119 / 0.095 | 7.26 / 6.03 / 5.62 |
| | DBSN [43] | 33.59 / 0.9389 / 0.060 | 6.57 / 5.48 / 5.54 |
| | R2R [27] | 32.21 / 0.8807 / 0.117 | 5.63 / 5.63 / 4.99 |
| | Neighbor2Neighbor [11] | 32.82 / 0.9275 / 0.087 | 6.47 / 5.86 / 5.33 |
| | Blind2Unblind [41] | 33.30 / 0.9380 / 0.061 | 5.28 / 5.22 / 5.04 |
| Ours | SelfIR | 34.51 / 0.9440 / 0.053 | 5.48 / 5.83 / 4.86 |

Table 3: Results of deblurring with clear images and noisy images as the supervision.

| Supervision Information | Gaussian $\sigma \in [5/255, 50/255]$ PSNR↑ / SSIM↑ / LPIPS↓ | Poisson $\lambda \in [5, 50]$ PSNR↑ / SSIM↑ / LPIPS↓ | Sensor Noise [4] PSNR↑ / SSIM↑ / LPIPS↓ |
|---|---|---|---|
| Clear Images | 28.24 / 0.8561 / 0.191 | 28.24 / 0.8561 / 0.191 | 28.14 / 0.8547 / 0.162 |
| Noisy Images | 28.29 / 0.8578 / 0.190 | 28.23 / 0.8563 / 0.191 | 28.16 / 0.8545 / 0.164 |

Table 4: Ablation study of auxiliary loss on Gaussian noise.

| | Neighbor2Neighbor [11] PSNR↑ / SSIM↑ / LPIPS↓ | SelfIR PSNR↑ / SSIM↑ / LPIPS↓ |
|---|---|---|
| w/o $\mathcal{L}_{aux}$ | 34.29 / 0.9271 / 0.085 | 35.65 / 0.9492 / 0.080 |
| w/ $\mathcal{L}_{aux}$ | 34.45 / 0.9307 / 0.093 | 35.74 / 0.9499 / 0.076 |

## 5 Ablation Study

### 5.1 Feasibility of Deblurring with Noisy Images

In order to verify the feasibility of taking noisy images as the supervision of deblurring in Sec. 3.1, we replace the clear supervision of Baseline$_\mathcal{B}$ with noisy images of different noise distributions, and the results are shown in Tab. 3. It can be seen that taking noisy or clear images as the supervision leads to similar performance on the deblurring task, which confirms that the noisy images can be an alternative of clear images to supervise the deblurring task.

### 5.2 Effect of Auxiliary Loss

In order to evaluate the effect of auxiliary loss $\mathcal{L}_{aux}$ in Eqn. (9), we add $\mathcal{L}_{aux}$ to the loss terms of self-supervised denoising method Neighbor2Neighbor [11]. As shown in Tab. 4, we obtain 0.16 dB PSNR gain against the baseline Neighbor2Neighbor [11]. The result indicates that blurry images can provide some auxiliary supervision information and improve performance for self-supervised denoising. When removing $\mathcal{L}_{aux}$ from our SelfIR, the PSNR dropped by 0.09 dB. Since the long-exposure image is taken directly as input, the loss term may be less effective, but is still beneficial.

Table 5: Ablation study on different weights ($\lambda_{reg}$ values) of regularization loss.

| $\lambda_{reg}$ | PSNR↑ / SSIM↑ / LPIPS↓ |
|---|---|
| 0 | 35.20 / 0.9473 / 0.097 |
| 1 | 35.64 / 0.9492 / 0.082 |
| 2 | 35.74 / 0.9499 / 0.076 |
| 4 | 35.72 / 0.9496 / 0.075 |
| 8 | 35.73 / 0.9497 / 0.072 |

Table 6: Ablation study on different weights ($\lambda_{aux}$ values) of auxiliary loss.

| $\lambda_{aux}$ | PSNR↑ / SSIM↑ / LPIPS↓ |
|---|---|
| 0 | 35.65 / 0.9492 / 0.080 |
| 1 | 35.73 / 0.9498 / 0.078 |
| 2 | 35.74 / 0.9499 / 0.076 |
| 4 | 35.73 / 0.9499 / 0.076 |
| 8 | 35.67 / 0.9496 / 0.077 |

## 5.3 Effect of Different Loss Weights

We conduct ablation studies on different weighting hyper-parameters (*i.e.*, $\lambda_{reg}$ and $\lambda_{reg}$) for balancing regularization and auxiliary loss terms. The experiments are conducted on Gaussian noise in sRGB space. When varying one hyper-parameter, the other one is set to 2 by default. As shown in Tabs. 5 and 6, it can be seen that the sensitivity to $\lambda_{reg}$ and $\lambda_{reg}$ of SelfIR is acceptable.

## 6 Conclusion

The complementarity between long-exposure blurry and short-exposure noisy images not only improves the performance of image restoration, but also makes it possible to learn a restoration model in a self-supervised manner. Jointly leveraging the long- and short-exposure images, we present a self-supervised image restoration method named SelfIR. On the one hand, we take the short-exposure images as the supervision information for deblurring. On the other hand, we utilize the sharp areas in the long-exposure images as auxiliary supervision information in aid of self-supervised denoising. SelfIR combines these two aspects by a collaborative learning scheme, and makes the deblurring and denoising tasks benefit from each other. Experiments on synthetic and real-world images show the effectiveness and practicality of our proposed method.

## 7 Limitation, Impact, Etc

This work is still limited in assessing the results on real-world image pairs. Although no-reference IQA metrics are adopted, these may be too unstable to report the performance consistent with humans accurately. We will establish a real-world blurry-noisy pair dataset including high-quality GTs to solve this problem in the future work. As for societal influence, this work is promising to be applied to terminal devices (*e.g.*, smartphones) for obtaining better images under low-light environments. It has no foreseeable negative impact. Besides, the images used in this work are from natural or human scenes. The GoPro dataset used for synthetic experiments is public under CC BY 4.0 license. There is no personally identifiable information or offensive content in the experimental data.

## Acknowledgement

This work was supported by the Major Key Project of Peng Cheng Laboratory (PCL2021A12) and the National Natural Science Foundation of China (NSFC) under Grants No.s U19A2073 and 61871381.

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
