# Self-Supervised Image Restoration with Blurry and Noisy Pairs (Supplementary Material)

**Zhilu Zhang[1], Rongjian Xu[1], Ming Liu[1], Zifei Yan[1,*], Wangmeng Zuo[1,2]**
[1] Harbin Institute of Technology, Harbin, China;
[2] Peng Cheng Laboratory, China
cszlzhang@outlook.com, ronjon.xu@gmail.com,
csmliu@outlook.com, yanzifei@hit.edu.cn, wmzuo@hit.edu.cn

## A  Content

The content of the supplementary material involves:

- Formulations of synthetic noise in Sec. B.
- Detailed collection process of real-world dataset in Sec. C.
- Detailed architecture of restoration network in Sec. D.
- More qualitative results in Sec. E.

## B  Formulations of Synthetic Noise

**Gaussian Noise.** Sensor noise mainly includes shot and read noise, which are caused by photon arrival statistics biases and imprecise readout circuits, respectively. Gaussian noise is a good approximation of read noise and is widely used as the assumption for denoising. In the experiments, the synthetic Gaussian noise $\mathbf{N}_{\mathcal{N}}$ for the short-exposure image can be expressed as,

$$\mathbf{N}_{\mathcal{N}} \sim \mathcal{N}(\mathbf{0}, \sigma^2 \mathbf{E}) \tag{A}$$

where $\mathcal{N}(\cdot, \cdot)$ denotes Gaussian distribution, $\mathbf{E}$ is an identity matrix with ones on the main diagonal and zeros elsewhere. Following previous works [3, 9], the standard deviation $\sigma$ is chosen uniformly at random from $[5/255, 50/255]$ when color intensity values of images are normalized to $[0, 1]$.

**Poisson Noise.** During imaging, the sensor receives photons and performs the photoelectric conversion. The quantum nature of light leads to uncertainty in the number of electrons, which introduces shot noise in the image. The shot noise follows a Poisson distribution $\mathcal{P}$ over the number of electrons, thus we also experiment using the noisy image $\mathbf{I}_{\mathcal{N}}$ with Poisson noise, which can be expressed as,

$$\mathbf{I}_{\mathcal{N}} \sim \mathcal{P}(\lambda \cdot \mathbf{I})/\lambda, \tag{B}$$

where $\mathbf{I}$ denotes a clean image, $\lambda \in [5, 50]$ according to [3, 9].

**Sensor Noise.** Recently, Brook *et al.* [2] propose to approximate shot and read noise as a single heteroscedastic Gaussian noise. The noise is more complex but more realistic than homoscedastic Gaussian or Poisson noise, which can be written as,

$$\mathbf{N}_{\mathcal{N}} \sim \mathcal{N}(\mathbf{0}, \lambda_{read} + \lambda_{shot}\mathbf{I}). \tag{C}$$

---

*Correspondence author.

36th Conference on Neural Information Processing Systems (NeurIPS 2022).

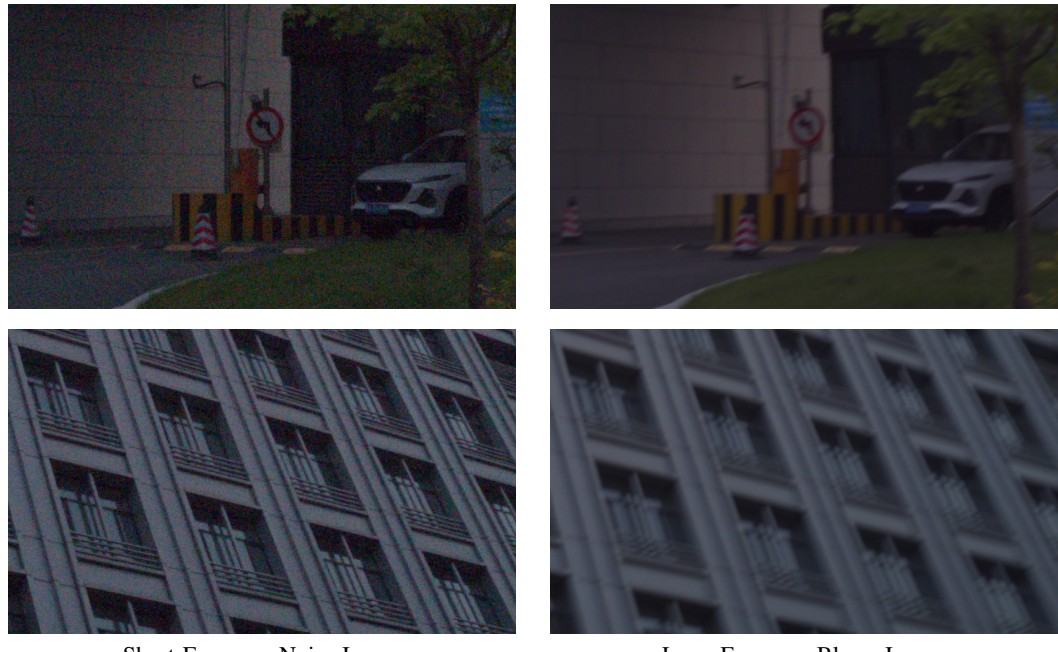

| Short-Exposure Noisy Image | Long-Exposure Blurry Image |

Figure A: Example image patches of our real-world blurry-noisy pairs.

where $\lambda_{read}$ and $\lambda_{shot}$ are determined by modeling the joint distribution of different shot/read noise parameter pairs in real-world raw images and sampling from that distribution. For SIDD [1] dataset, the sampling shot/read noise factors can be expressed as,

$$\log(\lambda_{shot}) \sim \mathcal{U}(\log(0.00068674), \log(0.02194856)),$$
$$\log(\lambda_{read}) \mid \log(\lambda_{shot}) \sim \mathcal{N}(1.85\lambda_{shot} + 0.3, 0.2^2). \tag{D}$$

where $\mathcal{U}(a, b)$ denotes a uniform distribution in the range $[a, b]$.

## C  Detailed Collection Process of Real-World Dataset

For obtaining real-world blurry-noisy image pairs, we utilize two Huawei P40 smartphones to capture a short-exposure noisy image and a long-exposure blurry image of the same scene, respectively. In the pair, the product of ISO and exposure time of two images remains the same, *i.e.*, the noisy image is captured with a high ISO and short exposure time, while the blurry one is captured with a low ISO and long exposure time. In order to make the captured image pair align as much as possible, the two smartphones take images side by side at the same time. Nonetheless, spatial misalignment still exists between the image pair, as the two camera lenses are not in the same position.

Therefore, we further exploit the optical flow network PWC-Net [8] to align the image pair. First, the raw-RGB pair is converted to sRGB space through camera image signal processing (ISP) pipeline. Then we calculate the optical flow between the sRGB image pair and use the optical flow to warp raw-RGB blurry image. Finally, the approximately aligned blurry-noisy raw-RGB pair can be obtained. Fig. A shows two examples of our real-world blurry-noisy image pairs.

## D  Detailed Architecture of Restoration Network

Fig. B shows the detailed architecture of restoration network. Two encoders are respectively deployed to the blurry and noisy input images for better domain-specific feature extraction, and then blurry and noisy features are fused in the decoder part. The architectures of encoder and decoder are the same with these of previous self-supervised denoising methods [3, 5, 6].

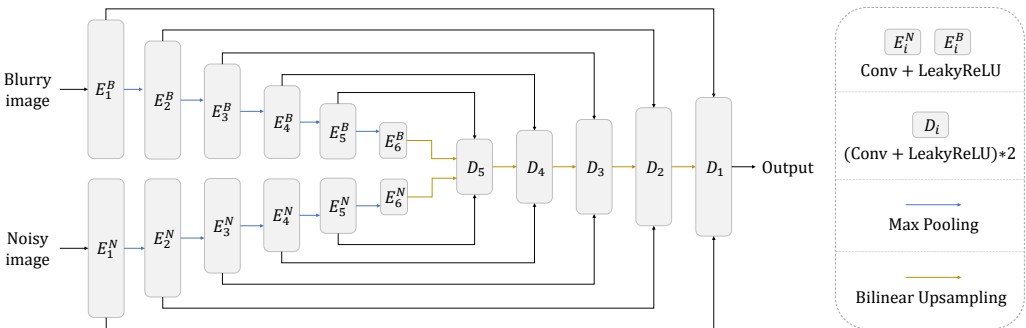

Figure B: Architecture of restoration network. $E_i^B$ and $E_i^N$ denote the $i$-th level of encoder for noisy image and blurry image, respectively. $D_i$ denotes the $i$-th level of decoder.

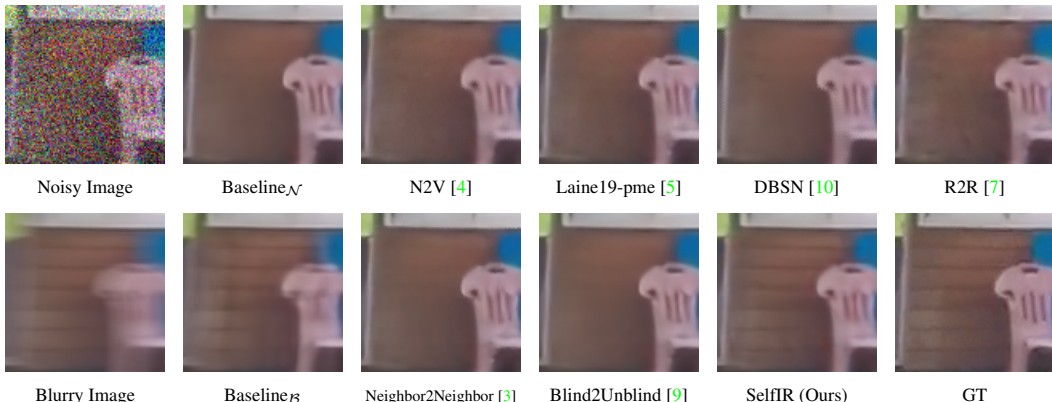

Figure C: Visual comparison on Gaussian noise. The lines and chair in our result are clearer.

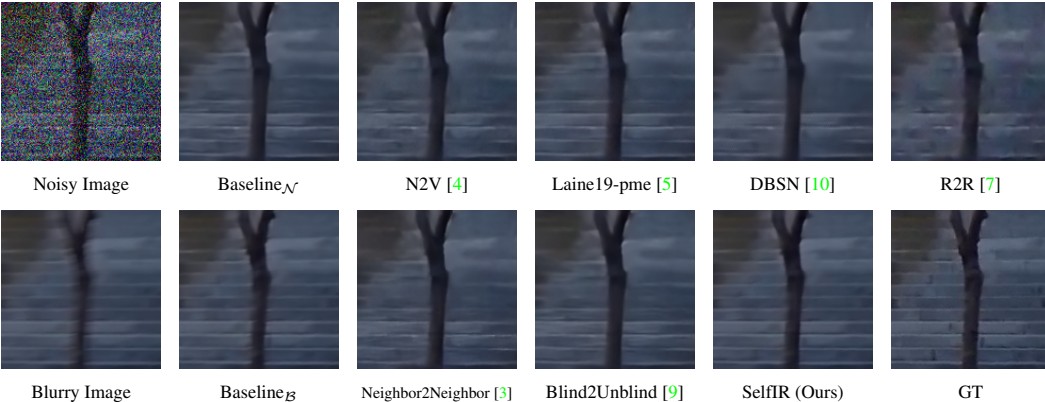

Figure D: Visual comparison on Gaussian noise. The edges of tree and steps in our result are sharper.

## E    More Qualitative Results

Figs. C and D show more qualitative results on Gaussian noise. Figs. E and F show more qualitative results on Poisson noise. The visual comparison on sensor noise [2] can be seen in Figs. G and H. The visual comparison on the real-world images can be seen in Figs. I and J. The results show our SelfIR performs favorably against the state-of-the-art self-supervised denoising methods, as well as the baseline supervised deblurring and denoising methods.

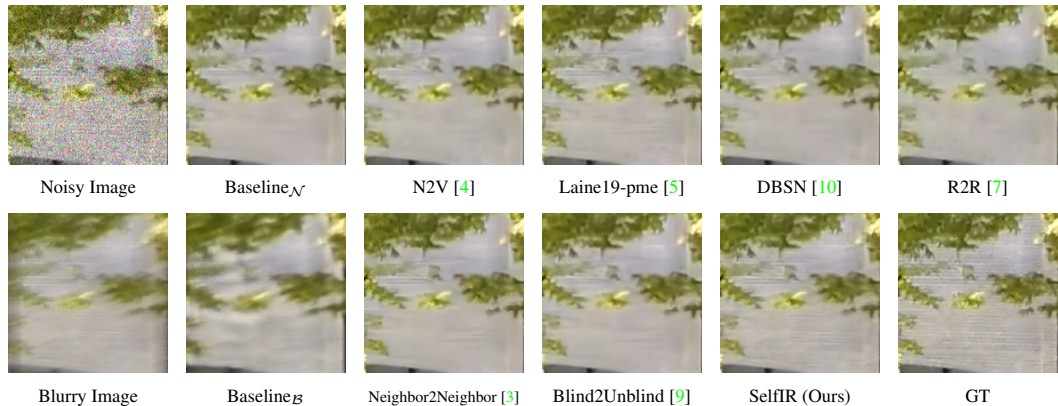

Figure E: Visual comparison on Poisson noise. Our SelfIR recovers more fine-scale textures.

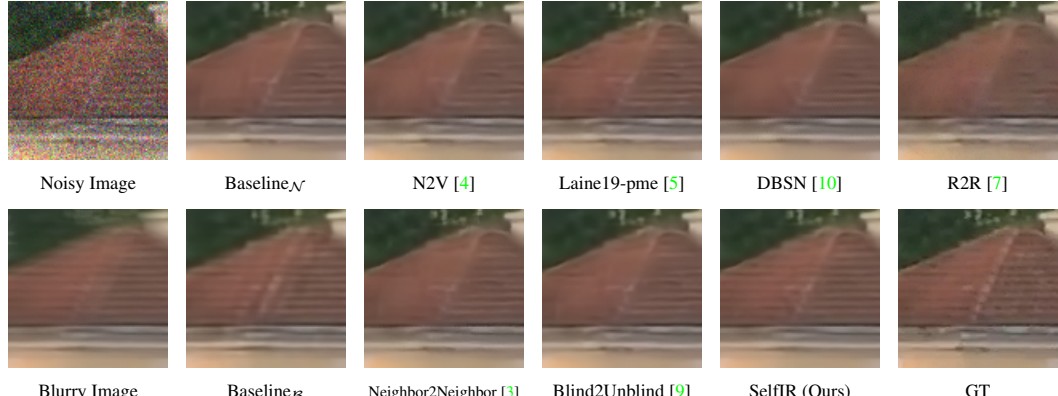

Figure F: Visual comparison on Poisson noise. The textures in our result are clearer and more regular.

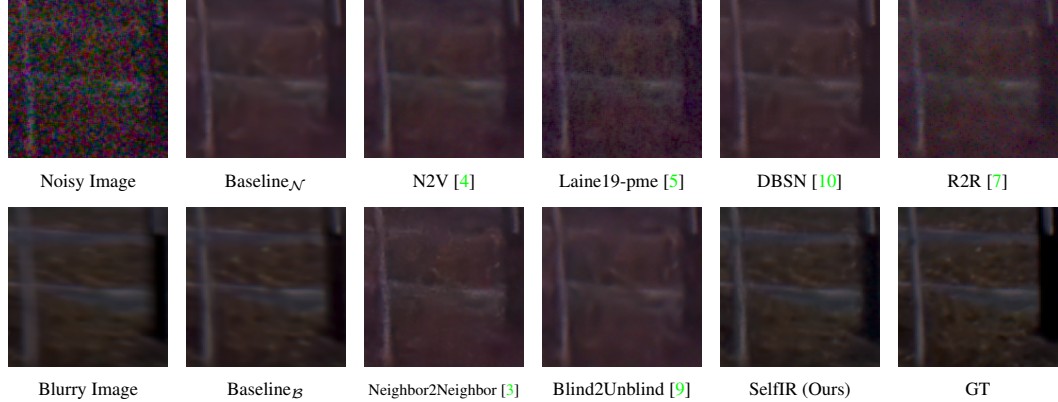

Figure G: Visual comparison on sensor noise. Our result is more photo-realistic and faithfully recovers the image color.

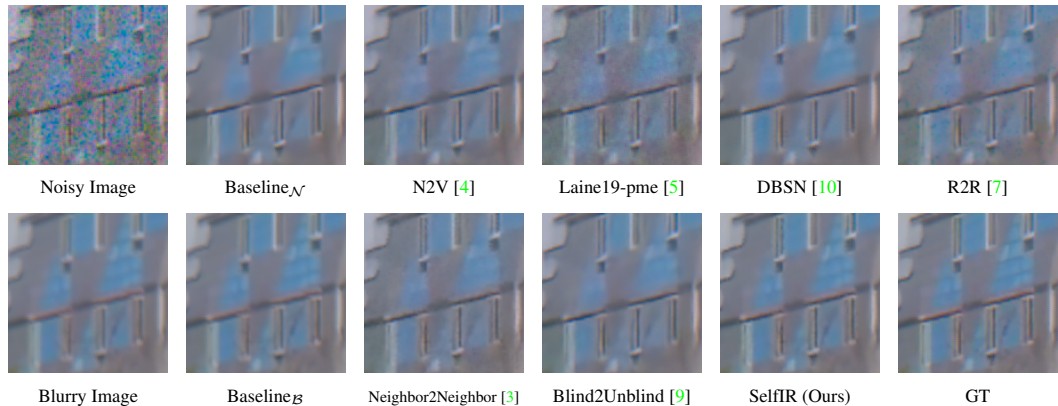

Figure H: Visual comparison on sensor noise. Our SelfIR recovers much more details.

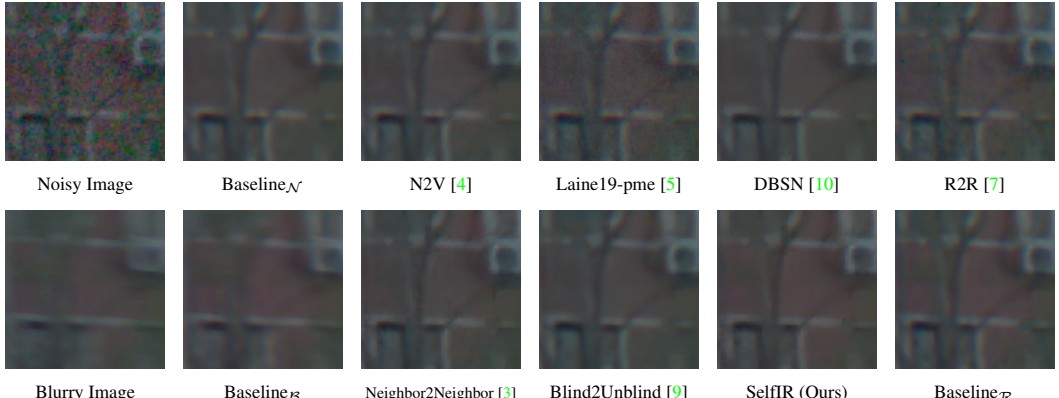

Figure I: Visual comparison on real-world images. Our result is clearer and more photo-realistic.

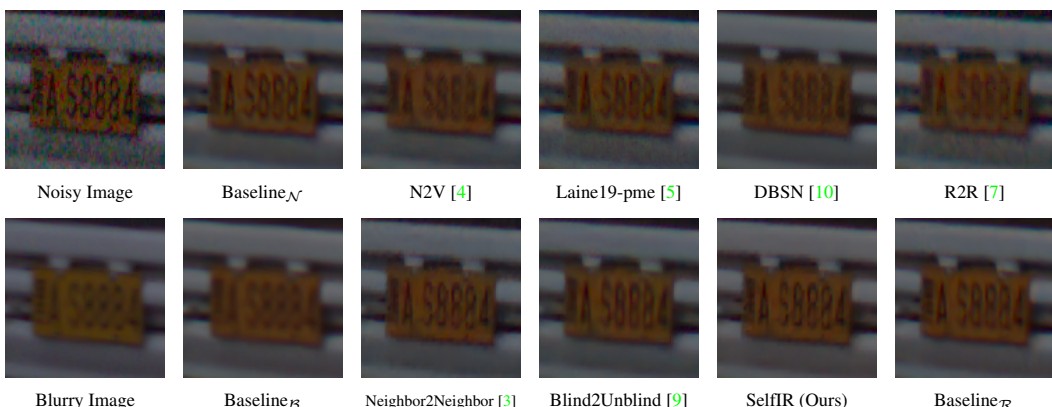

Figure J: Visual comparison on real-world images. The license plate in our result is clearer and sharper.