# OpenReview forum: "Self-Supervised Image Restoration with Blurry and Noisy Pairs"
_NeurIPS.cc/2022/Conference — NeurIPS 2022 Accept_

### Official Review · Reviewer_X2qR · 2022-06-18

**Rating:** 5
**Confidence:** 4
**Soundness:** 3 good
**Presentation:** 3 good
**Contribution:** 3 good

**Summary:**

This paper proposed a self-supervised image restoration method named ``SelfIR''. The basic idea is to work with a pair of short-exposure noisy and long-exposure blurry images. The complementary nature of blurry and noisy pairs make it possible to jointly utilize the auxiliary supervision information for self-supervised denoising. The reported experimental results are promising and convincing. SelfIR can achieve better performance than competing self-supervised denoising methods on synthetic datasets and comparable performance to N2N (ref. [15]) on real-world datasets.

**Questions:**

1. Is the proposed SelfIR "collaborative learning" or "self-supervised learning" or both (Line 170-171)? It seems to me Eq. (9) is collaborative learning since it involves the blurry-noisy image pair; while Eqs. (10)-(11) can be interpreted as self-supervised learning since it uses the pseudo-GT (output of restoration network D). Please correct me if I am wrong.
2. How is the sharp patch detected in Sec. 3.2 used by co-learning in Sec. 3.3?
3. Is there any risk of inconsistency between the patch-dividing strategy in Sec. 3.2 and sub-sampling operation in Sec. 3.3?
4. Do you assume the blur process with non-uniform kernels K is known in Eq. (1)? If Yes, this seems a strong assumption because nonuniform kernel estimation from real-world blurry images is a long-standing open problem.
5. Do you have visual quality comparison for denoising real-world raw sRGB images? Figs. 3-4 are for synthetic images only where supervised learning (baseline) is easy to obtain.

**Limitations:**

When does the proposed SelfIR fail? For example, Table 2 shows that the proposed method falls behind N2N [15] on PSNR and Blind2Unblind [37] on NIQE. Any discussion or explanation about the limitation of SelfIR?

**Strengths And Weaknesses:**

Strengths:
+ The idea of exploiting complementary nature of blurry and noisy image pairs is sensible. There is some novelty with the developed co-learning of deblurring and denoising in Sec. 3.3
+ The reported experimental results are mostly convincing and promising. Improved denoising results over Ref. [9] (CVPR'2021) and Ref. [37] (CVPR'2022) have been reported on synthetic sRGB images.

Weaknesses:
- Originality. I have found more clarification is needed for Sec. 3.1 and Sec. 3.3. How is Sec. 3.1 different from Noise2Nosie [15] and how is Sec. 3.3 different from Neighbor2Neighbor [9]? Many technical details such as Eqs. (4)-(6) and Fig. 2(b) have strong similarity with previous works. Even though the authors are inspired by those existing works, they need to highlight what are the new insights brought by this paper. I am afraid that the novelty of these two subsections is limited in the current version.
- Quality. I have two major concerns over Sec. 3.2 (denoising with long-exposure image). First, I have found that the strategy of dividing an image into non-overlapping patches ad-hoc. Even though "it is crucial to pinpoint the sharp areas in the long-exposure images", I don't understand why the authors take a patch-based (instead of a pixel-based) approach. Second, the use of SSIM in Eqs. (7)-(8) lacks substantial justification. Maybe it is based on heuristics but the quality of presentation for this paper is lacking, in my biased opinion.
- Clarity. I wish authors can clarify more about the self-supervised learning procedure in Sec. 3.3. The discussion about "complementarity" (line 169) is vague because authors did not explicitly discuss how the complementary nature was exploited by the design of co-learning method.
- Significance. The importance of this work seems to be to show the proposed SelfIR works better on real-world situations but the majority of the experimental results (especially visual comparisons) are for synthetic images only.

---

> ### Author Response · Authors · 2022-08-02
> **Response to Reviewer X2qR —— Part 2**
>
> 6. Is there any risk of inconsistency between the patch-dividing strategy in Sec. 3.2 and sub-sampling operation in Sec. 3.3?
>
> An experiment has been conducted to evaluate the impact of the sub-sampling operation on sharp areas detection.
> In the experiment, we only replaced $\mathbf{\tilde{I}}_\mathcal{N}$ in Eqn. (8) with $\hat{\mathcal{D}}(\mathbf{I}_\mathcal{B}, \mathbf{I}_\mathcal{N})$ for obtaining the mask, and then calculated $\mathcal{L}_\mathit{aux}(\mathcal{D}(\mathit{g}_1(\mathbf{I}_\mathcal{B}), \mathit{g}_1(\mathbf{I}_\mathcal{N}))), \mathit{g}_1(\mathbf{I}_\mathcal{B}))$.
> The results show that it has a similar performance to the method described in item 5.
>
>
> 7. Is it assumed that the blur kernel is known?
>
> No, we do not assume that the blur kernel is known.
> The real-world blur is generally caused by camera shake and object motion.
> Although some trajectory information of the camera shake can be obtained by the gyroscope, the object motion is very complex and it is difficult to give the motion blur kernel.
>
>
>
> 8. Visual quality comparison for real-world images
>
> Figures I and J in the supplementary material show visual comparison results for real-world images.
> It can be seen our results are clearer and more photo-realistic.
>
>
> 9. More explanations of limitations
>
> Neighbor2Neighbor [9] shows that the two images through neighbor sub-sampling from noisy sRGB image can be used as input-target pair to train a self-supervised denoising model.
> But for Bayer images in raw-RGB space, sampled pixels are not neighboring but sub-neighboring, as the neighbors of a pixel usually don't have the same color types as the pixel.
> It leads to a larger difference in the corresponding pixel-level content of the sampled images and does harm to performance.
> It is also the reason why Neighbor2Neighbor [9] and our SelfIR perform relatively poorer on synthetic raw noise than sRGB noise.
>
>
> For the evaluation of real-world images, due to the lack of ground-truth images, we have to utilize no-reference image quality assessment (IQA) metrics ($i.e.$, NIQE [19], NRQM [17], and PI [2]) to make quantitative comparisons.
> However, NIQE and NRQM are sometimes unstable to report the performance accurately.
> We recommend mainly referring to the more stable metric PI which is the combination of NIQE and NRQM.
> In terms of the PI metric, our SelfIR achieves the best performance.
> Furthermore, as described in Sec. 7, we will build a real-world dataset with high-quality ground-truths for improving experimental assessments.

---

> ### Author Response · Authors · 2022-08-02
> **Response to Reviewer X2qR —— Part 1**
>
> We thank the reviewer for the valuable comments and suggestions.
> We appreciate the reviewer's questions and hope our responses could address the concerns.
>
>
> 1. Originality and novelty (difference from Noise2Noise [15] and Neighbor2Neighbor [9])
>
> We kindly remind the reviewer that our main contribution is to leverage the nature of blurry and noisy image pairs to learn a self-supervised model and improve restoration performance.
>
> Noise2Noise [15] demonstrates that the denoising model can be trained with a noisy-noisy pair when the noise in the target image is independent of that in the input image.
> We present that the deblurring model can be supervised with a same-scene noisy image, which is not mentioned by Noise2Noise.
> We further demonstrate it through experiments in Table 3, which show that taking noisy or clear images as the supervision leads to similar performance on the deblurring task.
>
> In terms of Neighbor2Neighbor [9], we directly adopt its conclusion that the sub-sampled $g_1(I_N)$ and $g_2(I_N)$ images are content approximate and noise independent.
> That is to say, $g_2(I_N)$ can be as the supervision for denoising $g_1(I_N)$.
> Combined with the conclusion of Sec. 3.1 (Deblurring with Noisy Image), $g_2(I_N)$ can be also as the supervision for deblurring $g_1(I_B)$.
> Thus, we can input $g_1(I_N)$ and $g_1(I_B)$ together into the model, $i.e.$, $\\{g_1(I_N), g_1(I_B)\\}$, and take $g_2(I_N)$ as supervision information to perform self-supervised learning (see Eqns. (10) and (11)).
> In addition, the sharp areas in $g_1(I_B)$ can also provide auxiliary supervision for the model learning and the performance can be further improved.
>
>
>
> 2. More explanations of Sec. 3.2 (Denoising with Long-exposure Image)
>
> First, it is difficult to detect whether a single pixel is blurred or not.
> In other words, to build the mask of sharp areas at the pixel level, the pixel and its surrounding pixels are both required to form a patch to determine whether it is blurred or not.
> Therefore, we have tried dividing the image into overlapping patches to obtain a pixel-based mask.
> It shows a similar performance to the non-overlapping patch-based one, but brings more running time and GPU memory.
> Taking a trade-off between performance and efficiency, we adopt the non-overlapping patch-based way.
>
> Second, the SSIM term in Eqn. (8) is mainly used to detect the ringing artifact areas (as shown in Figure 1(c)) in blurry images.
> The variance of these areas is even larger than that of the denoised image.
> So it is easy for these areas to be mistaken as sharp areas when only variance is used.
> We have conducted an experiment that removes the SSIM term in Eqn. (8).
> The PSNR metric drops by 0.1 dB, which shows the effectiveness of the SSIM term.
>
>
> 3. Clarity and more explanations of complementarity
>
> The complementarity between blurry and noisy images lies in two aspects:
> (1) The noisy image can provide supervision information for deblurring (Sec. 3.1), and the blurry image can also provide auxiliary supervision for denoising (Sec. 3.2).
> So we can learn to deblur or denoise without ground-truths.
> (2) Better restoration performance can be obtained when jointing two images rather than learning deblurring or denoising task individually.
> So we further design the self-supervised restoration model that takes two degraded images as input and learns from both tasks (Sec. 3.3).
> In revision, we will reorganize Sec. 3.3 for a clearer statement.
>
>
>
> 4. Collaborative learning and self-supervised learning
>
> In Secs. 3.1 and 3.2, we introduce the deblurring subtask and denoising subtask individually for clarity.
> From the perspective of these two subtasks, it is collaborative learning since the information from the other is utilized.
> When we combine them into the proposed framework in Sec. 3.3, the task of this work is image restoration (deblurring and denoising are subtasks of it), and we can regard it as self-supervised learning.
>
>
>
> 5. How is the sharp patch detected in Sec. 3.2 used by co-learning in Sec. 3.3?
>
> In Sec. 3.3, for obtaining mask $\mathit{m}^\mathit{n}$, we replace $\mathbf{I}_\mathcal{B}$ and $\mathbf{\tilde{I}}_\mathcal{N}$ in Eqn. (8) with $\mathit{g}_1(\mathbf{I}_\mathcal{B})$ and $\hat{\mathcal{D}}(\mathit{g}_1(\mathbf{I}_\mathcal{B}), \mathit{g}_1(\mathbf{I}_\mathcal{N}))$, respectively.
> Note that $\hat{\mathcal{D}}$ has the same parameters with $\mathcal{D}$ but has no gradient for back-propagation.
> Then we calculate $\mathcal{L}_\mathit{aux}(\mathcal{D}(\mathit{g}_1(\mathbf{I}_\mathcal{B}), \mathit{g}_1(\mathbf{I}_\mathcal{N}))), \mathit{g}_1(\mathbf{I}_\mathcal{B}))$ according to Eqn. (9) for obtaining the auxiliary loss.

---

### Official Review · Reviewer_hPxE · 2022-07-12

**Rating:** 5
**Confidence:** 4
**Soundness:** 2 fair
**Presentation:** 3 good
**Contribution:** 3 good

**Summary:**

This paper proposes a self-supervised technique for image restoration from a blurry and a noisy image pair. By observing that each task can be trained from the other low-quality image, the authors propose a method that brings sharpness from the noisy image and the cleanliness from the blurry image.
However, I’m not sure if this method is a true self-supervised method. The training objective seems to be recovering the real noisy image from the blurry and (synthetic) augmented noisy image pair.


**Questions:**

L158 The ghost artifact has been pointed as imperfect blurry image in the following paper:

S. Nah et al., “NTIRE 2019 Challenge on Video Deblurring and Super-Resolution: Dataset and Study,” CVPRW 2019

They interpolate the frames before averaging the sharp (& noisy) images to remove the ghost artifact.
Following similar approaches may benefit finding sharp regions from equation (8).


**Limitations:**

Limitations described in the paper.

**Strengths And Weaknesses:**

[Strengths]

While there have been attempts to use the blurry and the noisy images to achieve better-quality images, this method is novel in terms of self-supervised design. The self-supervised learning process builds on top of the observations in the nature of deblurring and denoising. The experimental results look promising given my concerns are resolved.

[Weaknesses]

- Related Works

I think the authors should also refer to previous burst image denoising and burst image deblurring works.
The authors mainly describe self-supervised image denoising papers only. (+ 1 self-supervised deblurring paper)

- L196

From the definition in the introduction (L23-33) the blurry-sharp image pair in GoPro dataset is I_B and I_N pair (not I) as the sharp image is taken with a short exposure. The only difference from the assumption is that the GoPro dataset was captured in a bright environment where 240 fps video capturing is possible, however, noise exists and image quality is low.
The authors may augment the sharp & noisy image with more aggressive noise for training purposes (L196-199), however, the sharp image should not be considered a ground-truth.

Otherwise, the proposed method is not a true self-supervised method as I_N is generated from I in the training process. Without I (which is considered GT here), training is impossible.

The authors should at least show the proposed method could achieve higher-quality image D(I_B, I) which is cleaner and sharper than I.

In Figures I and J, is the noise in the noisy image real or synthetic? Also, how could the blurry and the real image be well-aligned?

- Basic setup

Are the blurry and the noisy image pair assumed to be taken 1) at the same time from different cameras or 2) at different times from a single camera?
For 1), the pair should be stereo images with disparity and 2) should have spatial offset between the pair due to motion.
However, training from a synthetic GoPro dataset has well-aligned image pairs. Does training with the aligned data generalize to realistic-scenario images?


- typos

L141 without than accept a shoddy option -> without (sharp areas?) than to accept a shoddy option

L147 patches.For -> patches. For

L172, 178 sub-sampling images -> sub-sampled images

L194 Hero 4 Black -> Hero4 Black

---

> ### Author Response · Authors · 2022-08-02
> **Response to Reviewer hPxE**
>
>
> We thank the reviewer for the valuable comments and suggestions.
> We appreciate the reviewer's questions and hope our responses could address the concerns.
>
>
> 1. More explanations on the GoPro dataset
>
> First, we kindly remind the reviewer that the definition in the introduction (L23-33) only applies to low-light photography.
> The GoPro dataset is captured in a bright environment.
> The observation that the short exposure image is noisy does not apply to GoPro dataset, as the low ISO can be set in bright environments.
>
> Second, even with some mild noise $\mathbf{N}$, it does not affect performance evaluation when taking sharp images of the GoPro dataset as ground-truths.
> We can denote the sharp image from GoPro as
>
> $\mathbf{I}_{GoPro}=\mathbf{I}+\mathbf{N},$
>
> where $\mathbf{I}$ is the ideal clean and sharp image.
> When calculating MSE (the intermediate result for obtaining PSNR) between the model output $\mathbf{\hat{I}}$ and $\mathbf{I}_{GoPro}$,
>
> $[\mathbf{\hat{I}}-\mathbf{I}_{GoPro}]^2=(\mathbf{\hat{I}}-\mathbf{I})^2-2(\mathbf{\hat{I}}-\mathbf{I})\mathbf{N}+\mathbf{N}^2.$
>
> Similar to Eqn. (5), the expectation of the second term is zero and that of the third one is constant.
> Thus, the relative order of the average PSNR index for the methods will not be affected.
> In fact, the ground-truth of many denoising datasets ($e.g.$, SIDD [1]) is obtained by averaging multiple images, where the noise is not fully eliminated as well.
>
>
> 2. More explanations of self-supervised learning
>
> For synthetic experiments, we add noise into sharp images of the GoPro dataset for generating noisy images.
> The sharp images are only used to synthesize noisy data, but not to calculate the loss function during training.
> Such operations are the same as previous self-supervised denoising methods ($e.g.$, N2V [11], Laine19 [13], DBSN [38], R2R [23], Neighbor2Neighbor [9], and Blind2Unblind [37]).
> Following these methods, our SelfIR can also be viewed as a self-supervised method.
>
> In addition, no ground-truths are used for  experiments on the real-world dataset.
> It further illustrates the self-supervised nature of our method.
>
>
> 3. Experiments on real-world dataset
>
> For real-world dataset acquisition, we take blurry and noisy images concurrently using two adjacent cameras in this work.
> Spatial misalignment indeed exists between the image pair.
> Therefore, we exploit the optical flow network PWC-Net [8] to align the image pair.
> Please see the detailed descriptions in Section C of the supplementary material.
>
> For evaluation on the real-world testing set, we use the real-world training set to fine-tune the model which is pre-trained with synthetic sensor noise in a self-supervised manner.
> Other self-supervised models to be compared are also trained in this way.
> Figures I and J show visual comparison results on real-world images, and the noise in the noisy images is real.
>
> Finally, we would like to point out that our main contribution is to exploit the complementarity of long- and short-exposure images for self-supervised image restoration and improving  performance.
> For the misalignment issue, on the one hand, notable progress has been made in optical flow estimation and this work utilizes them directly.
> On the other hand, another way for dataset acquisition can avoid the issue.
> We can take dozens or even hundreds of burst images with a camera, where the middle frame is taken as the short-exposure image, and the average of all frames is taken as the long-exposure one.
> Please see the detailed descriptions in `Response to Reviewer njZ6 - 1. Feasibility of real-world dataset acquisition'.
>
>
> 4. Related works and typos
>
> Many thanks.
> We will add some burst image denoising and burst image deblurring methods in related works.
> And we will fix these typos in the revision.
>
>
> 5. About the ghost artifacts
>
> Thanks for reminding the paper, we will cite this paper in the revision.
> However, we would like to leave the synthetic method unchanged to be consistent with the real situation, since ghost artifacts also exist in real-world blurry images (especially in areas with flickering lights).

---

> > ### Comment · Reviewer_hPxE · 2022-08-09
> > **Response to rebuttal**
> >
> > I thank the authors for the detailed comments.
> > Here are my follow-up comments.
> >
> > 1. GoPro dataset
> >
> > The authors of GoPro dataset admit the quality limitation in their follow-up paper:
> > S. Nah et al., "NTIRE 2019 Challenge on Video Deblurring and Super-Resolution: Dataset and Study," CVPRW 2019
> >
> > One of the main purposes of the REDS dataset was to improve the quality of reference images as quality limitations are prevalent in high-speed videos.
> > Their presentation material explains this issue in detail: https://drive.google.com/file/d/13F6UEyBDFGTiFDyxqLzrPiq4Y2-8BKQE/view?usp=sharing
> >
> > Besides,I agree with the authors that the proposed method works even with the inherent noise and I don't deny that.
> > What I wanted to deliver is that GoPro should not be considered defect-free and what is written in the paper may confuse readers.
> >
> > 2. Self-supervision
> >
> > I understand what the authors mean, however, I'm still reluctant to refer to this method as self-supervised, at least for the synthetic dataset experiments.
> > A more precise explanation would be that this method applied to real-world datasets in a self-supervised manner.
> >
> > 3. Real-world datasets
> >
> > Thanks for the explanation.
> >
> > 4. Ghost artifacts
> >
> > Authors may also want to refer to additional works:
> >
> > * P. Wieschollet et al., "Learning blind motion deblurring," ICCV 2017
> > * S. Zhou et al., "DAVANet: Stereo Deblurring with View Aggregation," CVPR 2019

---

> > > ### Author Response · Authors · 2022-08-09
> > > **Response to Reviewer hPxE**
> > >
> > >
> > > Thanks for your more detailed comments.
> > >
> > > 1. GoPro dataset
> > >
> > > Thank you for your more detailed explanation.
> > > We acknowledge that the sharp images in the GoPro dataset are not defect-free, and will modify the relevant descriptions in the revision.
> > >
> > >
> > > 2. Self-supervised learning
> > >
> > > To begin with, we respect different opinions about self-supervised learning.
> > > However, following the setting of synthetic experiments in previous self-supervised image denoising methods, only the corrupted images (with a single degradation, $i.e.$, noise) are utilized for model training.
> > > Similarly, for the image restoration task in this paper, only the corrupted images (with multiple degradations, $i.e.$, noise and blur) are utilized for model training.
> > > In other words, even for synthetic experiments, the ground-truth images are not involved in the training phase.
> > > Therefore, it can be regarded as self-supervised learning.
> > > We hope this response could address your concerns.
> > >
> > >
> > > 3. Ghost artifacts
> > >
> > > Many thanks. We will add these works in the revision.

---

### Official Review · Reviewer_njZ6 · 2022-07-20

**Rating:** 6
**Confidence:** 3
**Soundness:** 4 excellent
**Presentation:** 4 excellent
**Contribution:** 2 fair

**Summary:**

This paper considers the problem of taking high quality clean images given the difficulty of the tradeoff between obtaining blurry images due to long exposure times, and noisy images from small exposure time. A typical way to handle the tradeoff is to find a compromise between the two, then use a debluring or denoising model to fix up the image (the models trained using supervision or self-supervision). Instead this paper proposes to make use of both a blurry and a noisy image, taking advantage of the fact that the blurry image will have very little noise, and the noisy model very little blur in order to co-train *both* a debluring and a denoising model. Using these two models the authors Provide a method for producing a single clean image given a blurry and a noisy image.


**Questions:**

See previous section.

**Limitations:**

Yes.

**Strengths And Weaknesses:**


The main contribution of this work seems to be the idea that noisy and blurry images contain complimentary information, that, if both images are collected, can be used to train both a debarring and a denoising model. Using both of theses models to restore images then leads to good clean images. This message seems clear and intuitive, and the method a useful contribution. Too I am happy to lean towards acceptance. But there is an important major caveat. It. Is non-trivial, even infeasible to take both a long- and short-expoosure image of the same scene. The authors solution is to use two cameras side by side taking photos simultaneously. So I am quite worried about the general availability of the type of data required to use this method.

---

To summarize some points:
- The paper is clearly written and highly accessible.
- The idea of using the complimentary information in noisy vs blurry images is natural and intuitive
- Strong quantitative and qualitative results: the proposed method seems to improve over existing  supervised & self-supervised denoising methods (& more).

---

Some questions:
- The real-world dataset only contains 61 images, only 30 of which are used for training. This is clearly a really small dataset. I would really like to hear from the authors on their thoughts about this data issue, and how it affects the viability of the joint-training method.
- Could other intermediate exposure time images also be incorporated? So instead of just having very short and relatively long exposures, also include medium length exposures.
- The overall loss includes two weighting hypermarameters (Eqn 12). How sensitive is the method to the choice of these hyper parameters?
- Recent works in diffusion modeling have found that when denying a sample it is better to parameterize the denoising model to *rpedict the noise* instead of directly predicting the clean sample. Did you at all look into this? It could either 1) improve the current method, or 2) provide interesting counter-evidence to the generality of the observation from diffusion modeling.

---

A disclaimer: I do not keep up to date with the recent literature in image restoration, so potentially will not be aware of related work.

---

> ### Author Response · Authors · 2022-07-31
> **Response to Reviewer njZ6**
>
> We thank the reviewer for the valuable comments and suggestions.
> We appreciate the reviewer's questions and hope our responses could address the concerns.
>
> 1. Feasibility of real-world dataset acquisition
>
> The real-world data can be obtained in the following three ways:
> (1) taking two images concurrently with two adjacent cameras;
> (2) taking two images one-by-one with a single camera;
> (3) taking dozens or even hundreds of burst images with a camera, where the middle frame is taken as the short-exposure image, and the average of all frames is taken as the long-exposure one.
>
> Since we have no access to the camera's internal control, we choose the first way, which is the most feasible approach for us.
> But for manufacturers' practical applications, the third one may be more suitable.
> In this way, the averaged (long-exposure) image over multiple noisy images has negligible noise, as the noise in the burst images is random and generally zero-mean.
> Moreover, there is no misalignment between the long- and short-exposure images using the third way.
>
>
> 2. More explanations on our real-world dataset.
>
> The main purpose of collecting the dataset is to demonstrate the effectiveness of SelfIR on real-world images.
> Limited by time and equipment, the scale of the dataset is indeed small.
> In order to avoid insufficient evaluation, we take as many ($i.e.$, half of) images as possible for testing.
> In order to avoid insufficient training, we first pre-train the model with synthetic sensor noise (in a self-supervised manner) and then fine-tune it on the collected training images.
> After fine-tuning, the PI [2] metric on the collected real dataset is improved by 0.55, we will add the result in the revision.
>
> In addition, it is also our follow-up goal to build a larger-scale open dataset with ground-truths, and we hope it could better facilitate the development of this task.
>
> 3. Could other intermediate exposure time images also be incorporated?
>
> We believe that it has the potential to further boost the model performance, since the intermediate exposure time images may bring more useful and richer information.
> However, it is non-trivial to directly extend our framework to involve these images (both noisy and blurry) in a self-supervised manner.
> Therefore, we would like to reconsider the framework and explore it in future work.
>
>
> 4. Weighting hyperparameters of loss terms
>
> The weighting hyperparameters $\lambda_{reg}$ and $\lambda_{aux}$ are both set to 2 by default for experiments in sRGB space.
> Here, we have varied $\lambda_{reg}$ or $\lambda_{aux}$ to conduct experiments on Gaussian noise.
> The following two tables show the experimental results.
> The results show that the sensitivity to $\lambda_{reg}$ and $\lambda_{aux}$ of SelfIR is acceptable.
> We will add the results in the revision.
>
> |$\lambda_{reg}$| PSNR / SSIM / LPIPS    |
> | :----:   |  :----:  |
> | 0             | 35.20 / 0.9473 / 0.097  |
> | 1             | 35.64 / 0.9492 / 0.082 |
> | 2             | 35.74 / 0.9499 / 0.076 |
> | 4             | 35.72 / 0.9496 / 0.075 |
> | 8             | 35.73 / 0.9497 / 0.072 |
>
>
> |$\lambda_{aux}$| PSNR / SSIM / LPIPS    |
> | :----:   |  :----:  |
> | 0             | 35.65 / 0.9492 / 0.080 |
> | 1             | 35.73 / 0.9498 / 0.078 |
> | 2             | 35.74 / 0.9499 / 0.076 |
> | 4             | 35.73 / 0.9499 / 0.076 |
> | 8             | 35.67 / 0.9496 / 0.077 |
>
>
> 5. About the diffusion model
>
> We admitted that we currently know not so much about the diffusion model.
> To the best of our knowledge, most existing diffusion models in image restoration are learned in a supervised manner [a, b, c].
> So it still requires further explorations to achieve self-supervised image restoration with blurry and noisy pairs via diffusion models.
>
> [a] Saharia, Chitwan, et al. "Image super-resolution via iterative refinement." arXiv preprint arXiv:2104.07636 (2021).
>
> [b] Rombach, Robin, et al. "High-resolution image synthesis with latent diffusion models." CVPR. 2022.
>
> [c] Whang, Jay, et al. "Deblurring via stochastic refinement." CVPR. 2022.

---

> > ### Comment · Reviewer_njZ6 · 2022-08-08
> > **Rebuttal response**
> >
> > Thank you for the useful information provided in the rebuttal. Particularly I am happy with the explanation provided for the data acquisition. I do not have a deep understanding of photography so am not able to determine for myself how easy it is to modify a camera's internal control, but I am wiling to take your word for it that it would in principle be possible for those who do have more control over the camera mechanism to gather pairs of images according to the third method you described. On this basis I am cementing my assessment.

---

> > > ### Author Response · Authors · 2022-08-09
> > > **Response to Reviewer njZ6**
> > >
> > > Many thanks for your acknowledgment. We will revise the paper according to your comments.

---

### Meta-Review · Area_Chair_cT7h · 2022-08-28

**Recommendation:** Accept
**Confidence:** Certain

**Metareview:**

All three reviewers voted to accept the paper, and the detailed rebuttals from the authors helped to clarify reviewers' original concerns. One remaining concern from one of the reviewers is whether this method should be referred to as "self-supervised".  However, authors clarified that it is reasonable to consider this method self-supervised for real-world data. I am fine with therefore leaving "self-supervised" in the title.


**Award:**

No

---

### Decision · Program_Chairs · 2022-09-14

Accept